# The effect of increasing indoor ventilation on artificially generated aerosol particle counts

**Ashwin Johri** *

Collegiate School, Richmond, Virginia, United States of America

* ashwinjohri@icloud.com

## Abstract

The COVID-19 global pandemic has caused millions of infections and deaths despite mitigation efforts that involve physical distancing, mask-wearing, avoiding indoor gatherings and increasing indoor ventilation. The purpose of this study was to compare ways to improve indoor ventilation and assess its effect on artificially generated aerosol counts. It was hypothesized that inbuilt kitchen vents would be more effective in reducing indoor aerosol counts than opening windows alone. A fixed amount of saline aerosol was dispersed in the experimental area using a nebulizer under constant temperature and a narrow range of humidity. A laser air quality monitor was used to record small particle counts every 30 minutes from baseline to 120 minutes for four different experimental groups for each combination of kitchen vents and windows. The results of the study demonstrate that aerosol counts were lowest with the kitchen exhaust vents on. This study suggests that liberal use of home exhaust systems like the kitchen vents could achieve significantly more air exchange than open windows alone and may present an effective solution to improving indoor ventilation, especially during the colder months when people tend to congregate indoors in closed spaces. There were no safety concerns involved when conducting this experiment.

**Data Availability Statement:** All relevant data are within the manuscript and its Supporting Information files.

**Funding:** The author(s) received no specific funding for this work.

## Introduction

The COVID-19 pandemic caused by the novel coronavirus, SARS-CoV-2, was first identified in Wuhan, China in December 2019. Since then, it has rapidly spread globally, causing more than 200 million infections and more than four million deaths. The disease burden has been immense in the US with more than 37 million infections and more than 600,000 deaths [1]. COVID-19 is highly contagious and is believed to spread via droplet and aerosol transmission and efforts to mitigate the spread of COVID-19 have included physical distancing, masking around others, and limiting travel. Despite these recommendations, the surge of infections associated with indoor gatherings led to the realization that airborne transmission within households was occurring [2]. A combination of closed spaces, crowding, and presence of susceptible individuals resulted in a perfect viral "engine" and a surge in cases with super spreader events after indoor mass gatherings [3, 4]. Scientists have suggested outdoor meetings since outdoor environment is associated with lower risk of transmission of COVID 19 virus, but this is difficult to achieve during colder months of the year [5]. General guidance from the Centers

**Competing interests:** The authors have declared that no competing interests exist.

for Disease Control (CDC) suggested improving ventilation in homes but research did not reveal a direct comparison between various specific options in a single-family home [6].

Respiratory infections like influenza and COVID-19 are spread from transmission from person to person with infectious particles that are shed in respiratory secretions from the infected person and contracted by a healthy person who breathes in infectious particles from the air [7]. Respiratory droplets are larger particles produced by sneezing and coughing; they measure 5–100 microns and fall within 1–2 meters from the source [8]. Respiratory droplets can be infective if a person is in close proximity to the source or comes in contact with surfaces where these droplets settle [9]. Recently, more focus has been on respiratory aerosols which are particles suspended in the air that measure less than 5 microns that can remain airborne for periods of time and if inhaled, reach and invade the air sacs of the lung and cause pneumonia [10, 11]. Respiratory aerosols are generated with breathing, coughing, singing, sneezing, and speaking, and can remain suspended in the air for several hours, sometimes traveling distances of 23–27 feet [12–14].

Environmental factors and particle size of aerosols play a role in how long the infectious particle could be viable in a confined space [15]. Previous studies on viruses have suggested that relative humidity and ambient temperature play a role in the viability and transmission of the virus, with the virus being unstable at extremes of environmental temperature and humidity [16, 17]. It is believed that at higher relative humidity, the bioaerosol takes on more moisture and is likely to transform into a droplet that is larger in size and that settles quickly. With lower relative humidity, the bioaerosol can desiccate and can remain suspended or travel several feet based on air current and flow [18, 19]. Similar theories have been put forward in the transmission of the SARS-CoV-2 virus via aerosols as well. In the hospitals, COVID 19 patients are admitted to "airborne" isolation rooms which means that the room is undergoing a set number of air exchanges per hour to limit transmission to health care workers who are at high risk of contracting the virus from an infected patient and can subsequently become a source of infection as has been cited during super spreader hospital outbreaks [20]. The air from the room is exchanged through filters or is vented to the outdoors and the infectious aerosols are either absorbed in the filters or diluted in ambient air outdoors and have been used for treating patients with tuberculosis for years. Recommended air exchange per hour (ACH) in the hospitals is 6 for existing buildings and 12 for new buildings with recommendations to upgrade ACH closer to 12 with modifications where feasible [21].

Kitchen vents working as exhausts to the outside are known to exchange indoor air and improve air quality for indoor pollution like cooking fumes [22, 23]. General guidance from the CDC has included opening windows and using exhaust fans in home kitchens and bathrooms but direct comparisons have not been studied [6]. This study was performed to test the hypothesis that the kitchen vents will reduce the aerosol counts in a home setting at constant relative humidity and indoor temperature to a greater degree than ventilation with open windows.

## Materials and methods

The experimental trials were performed from October to December of 2020 in the combined kitchen and breakfast dining area of a single-family home with an open floor plan that had three 30x30 inch screened windows adjacent to the seating area (Fig 1). The middle window was used as a variable for this experiment. The kitchen stovetop (Kenmore Elite Downdraft Gas Cooktop model 790.31112111 36) and inbuilt ventilation system providing 525 cubic feet per min of airflow was located on a central kitchen island and 12 feet from the seating area where the experiment was set up (Fig 1). Dylos DC 1100 Pro air quality monitor (US patent

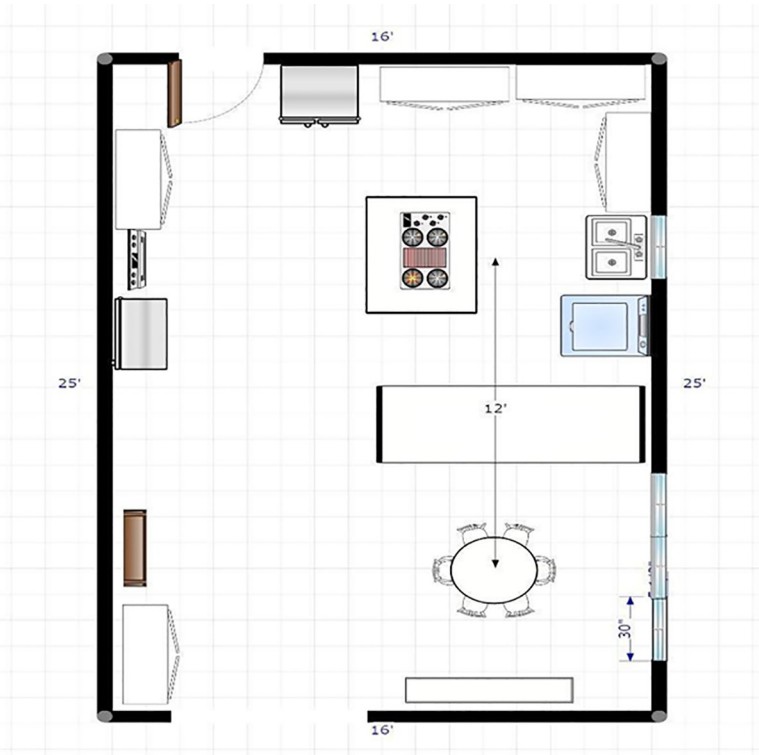

**Fig 1. Dimensions and layout of the space where experiment was conducted.**

8009290) with a laser counter capable of detecting small (0.5 microns) and large (2.5 microns) particles was used. Since it was noted that the particle counts went up with cooking, the experiment was performed at least 6 hours after the kitchen was used to allow the indoor particle counts to return to the baseline value confirmed by the particle counter. Four scenarios were tested and compared: group 1—vent on, window closed, group 2—vent off, window closed, group 3—vent on and window open, and group 4—vent off, window open. Aerosol generation was done using 3 ml of sterile saline nebulized into the seating area space using the Vios aerosol delivery system (Vios compressor and Pari nebulizer) for 20 mins (which is how long it took for complete nebulization of 3 ml of saline) Readings were taken at baseline, time 0 (when the nebulizer was switched off after 20 mins of aerosolization), 5 mins baseline (to allow for immediate stabilization of aerosol counts), 30, 60, 90, and 120 mins. Central heating was consistent throughout the experiment and the temperature kept constant at 70˚ F with the HVAC fan powered on throughout the experiment. An Accurite humidity monitor with a digital display was used to monitor the relative humidity and a Pure Enrichment room humidifier was used to keep the relative humidity in the desired range of 35% to 40%. A Benetech anemometer (model GM816) was used to measure airflow at the kitchen vent and the open window. All analyzed readings were taken at constant temperature (70˚ F) and a narrow range of relative humidity (35–40%).

The initial experiment was exploratory. Readings were taken with central heating off and varying relative humidity. Both small and large particle counts were higher when relative humidity was high and higher levels were detected for longer periods of time. Particle counts were also higher when the readings were taken immediately after cooking in the kitchen area. To minimize confounding variables, baseline particle count was measured to ensure counts

were near the expected baseline. Dylos particle counter capable of continuous recording was switched on. After ensuring particle counts were close to baseline (200–500 range), 3 ml of sterile saline was placed in the nebulizer cup and the compressor switched on for 20 minutes for all trials. Particle count was noted immediately after as "time 0". A new baseline allowing for immediate dispersion and stabilization of small particle count after 5 mins was noted as "5 mins baseline". Subsequently, 30, 60, 90, and 120 mins readings were obtained with kitchen vent on, window closed (group 1), vent off, window closed (group 2), vent on, window open (group 3) and vent off, window open (group 4) with 10 trials of each group and average calculated. The rate of decline in small particle size with time was compared between different groups. Exploratory readings were taken for group 5 as well which comprised of kitchen vent off, windows closed and air purifier with HEPA filter on but this group was not considered for comparative analysis because it did not accurately reflect air exchange or improvement in ventilation.

To confirm significant airflow was occurring, air flow was measured using a hand held anemometer at the kitchen vent which recorded an outgoing (exfiltration) air velocity of 6 m/sec to the outside. The open window recorded an incoming (infiltration) air velocity of 0.9 m/sec when the vent was not on and 1.6 m/sec when the kitchen vent was on, confirming an increase in infiltration occurring at the open 30 x 30 inches screened window next to the seating area resulting in more air exchange when the kitchen vent was running. Review of the user manual of the kitchen cooktop model revealed that the fan for downdraft unit was capable of moving 525 cubic feet of air per min vented through a duct to the outside. With room dimensions measured at a length of 25 feet, a width of 16 feet and a ceiling height of 9 feet, air exchange per hour (ACH) was calculated using the following formula: $ACH = \frac{CFM \times 60}{Area \times Ceiling\ Height}$ [24].

## Results

Data collected with the laser particle counter included small particle counts and large particle counts. Rate of decline with time in small particle counts, which was of primary interest as the closest simulation to an airborne infectious aerosol, was used for comparative analysis. The average particle count for all groups was calculated for baseline, time 0, 5, 30, 60, 90, and 120 mins (Table 1). Group 1 (vent on, window closed) showed a high rate of decline with 35%, 20%, 11%, and 8% of the small particles remaining at 30, 60, 90, and 120 minutes. Group 3 (vent on, windows open) showed the maximum decline with 34%, 18%, 11% and 6% at those time points. Group 2 (vent off, windows closed) had the slowest decline in small particle count with 75%, 56%, 35% and 27% recorded while Group 4 (vent off, window open) measurements were 62%, 46%, 34%, and 21% (Table 2). Although readings were obtained in Group 5 (vent off, window closed, HEPA filter on), this group was not used in comparative analysis as there was a distinct probability that the aerosol particle counts would be falsely detected as low because of dispersion while being blown through the filter and not because of increased ventilation in the room. Based on dimensions of the room (length 25 feet, width 16 feet, height 9

**Table 1.  Average small particle counts detected at different time points.**

| Groups | Baseline | 0 mins | Baseline 5 mins | 30 mins | 60 mins | 90 mins | 120 mins |
|---|---|---|---|---|---|---|---|
| Group 1 | 239.1 | 16868.6 | 13009.6 | 4513.2 | 2587.2 | 1470 | 1088.2 |
| Group 2 | 270 | 14706.7 | 12769.8 | 9607.8 | 7277 | 5782.7 | 3481.3 |
| Group 3 | 320.6 | 18862.8 | 13914 | 4773.1 | 2518.3 | 1660.2 | 967.5 |
| Group 4 | 289.4 | 15279.1 | 12892.9 | 7994.7 | 6054.7 | 4395.4 | 2784.1 |
| Group 5 | 287.8 | 17993 | 11667.4 | 4803 | 2680 | 1669.2 | 989.6 |

**Table 2. Percent values of small particle counts detected at different time points from 5 mins baseline.**

| Groups | 5 mins Baseline | 30 mins | 60 mins | 90 mins | 120 mins |
|---|---|---|---|---|---|
| Group 1 | 100% | 35% | 20% | 11% | 8% |
| Group 2 | 100% | 75% | 56% | 35% | 27% |
| Group 3 | 100% | 34% | 18% | 11% | 6% |
| Group 4 | 100% | 62% | 46% | 34% | 21% |

feet) and specifications of the downdraft fan (downdraft moving 525 cubic feet per minute), air exchange per hour (ACH) with the use of the kitchen vent was calculated and was found to be 8.75.

Small particle aerosol counts generated by a nebulizer were measured at different intervals in four experimental groups. The measurements were taken at constant temperature and comparable relative humidity maintained between 35–40%. Data recorded and analyzed suggested that the kitchen vent was more effective in lowering aerosol counts from time 0 to time 120 mins when compared to open window alone. Both kitchen vent and open window were better at lowering aerosol counts when compared to closed windows and vent switched off. The average values of small particle counts in different groups at baseline, time 0, 5, 30, 60, 90 and 120 minutes are shown in Figs 2–5. Comparison of percent value of small particle counts from baseline as detected at time 30, 60, 90, and 120 mins is shown in Fig 6 where it is apparent that the quickest decline occurred in groups 1 and 3, both of which involved actively operating kitchen vents.

## Discussion

This experiment supported the hypothesis that kitchen vents can work as an exhaust system resulting in air exchange and can reduce indoor aerosol counts more effectively in an indoor space with no added ventilation or with open windows. With the kitchen vent model as

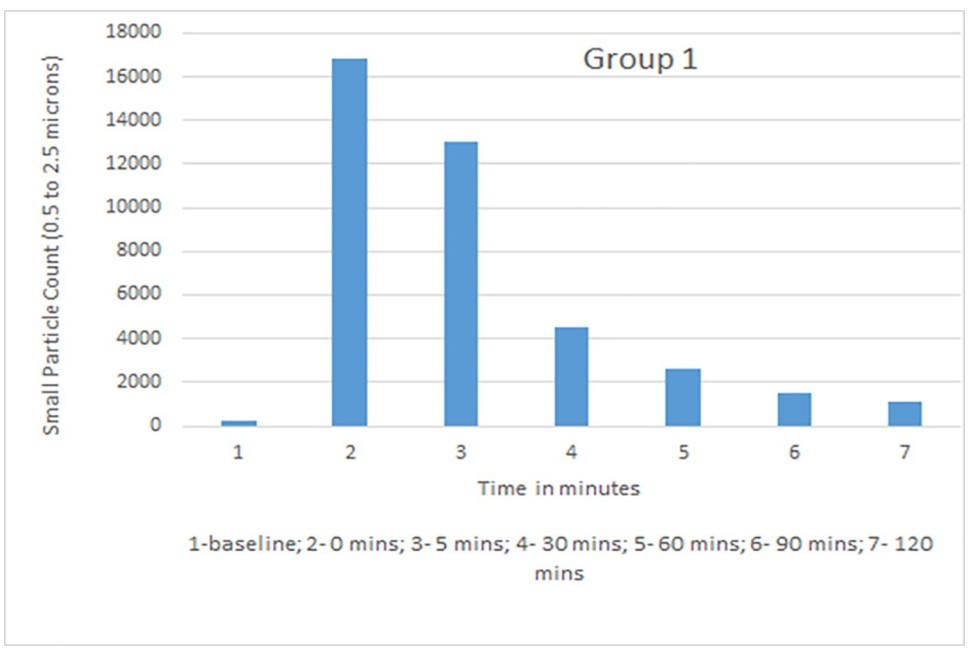

**Fig 2. Small particle counts for group 1 (vent on, window closed).**

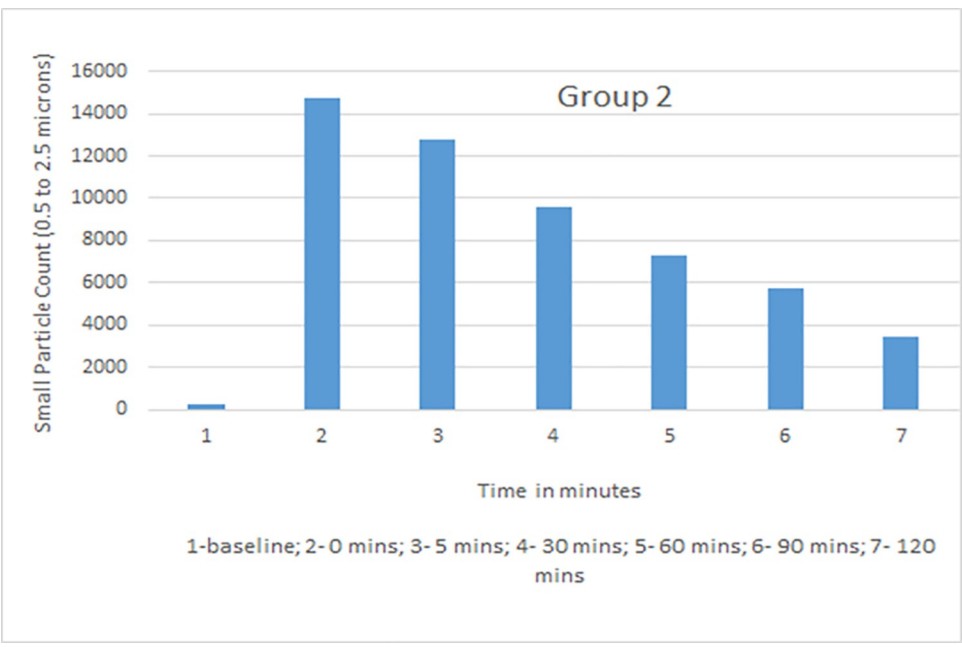

**Fig 3. Small particle count for group 2 (vent off, window closed).**

specified above and dimensions of the room, 8.75 air exchanges per hour were achieved. A one-way analysis of variance (ANOVA) test, after ensuring that the correct requirements were met, was used to determine statistically significant differences in the means of air particles cleared in each group (Fig 7). The determined p-value of the test, 0.000575, was far less than the standard significance level of 0.05, so there is significant support for the null hypothesis to

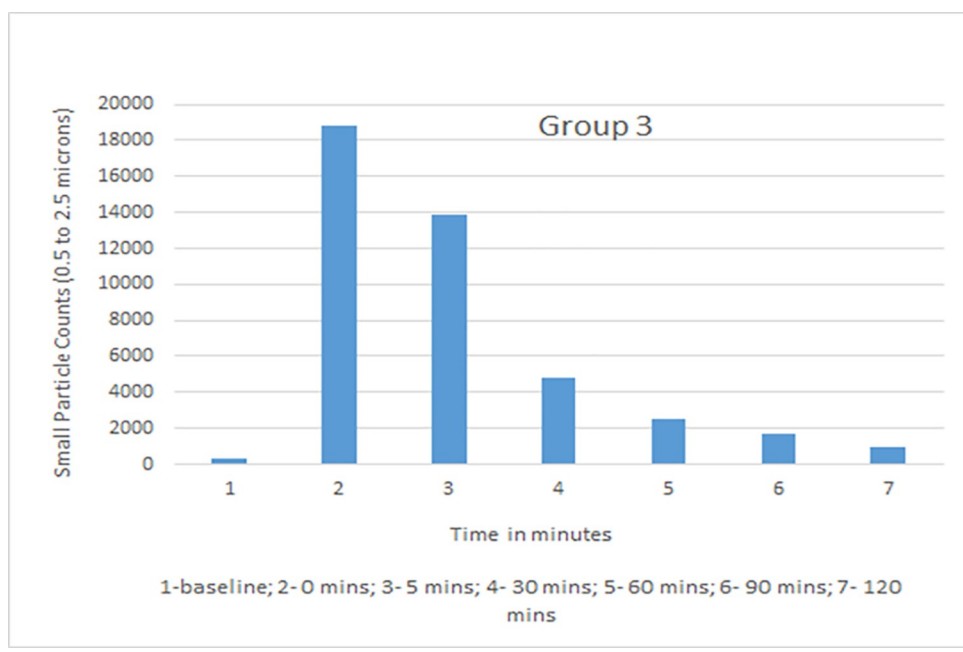

**Fig 4. Small particle count for group 3 (vent on, window open).**

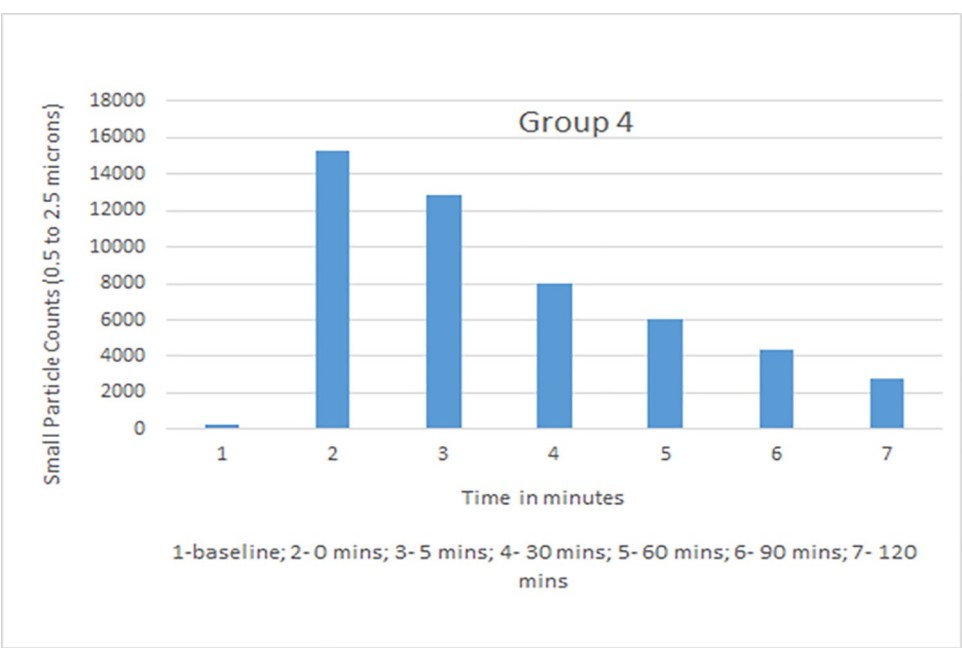

**Fig 5. Small particle count for group 4 (vent off, window open).**

be rejected, and it can be assumed that the means of each of the experimental groups analyzed are not the same.

The results from this observational study show that indoor ventilation can be enhanced by simple means using home exhaust systems. The absolute number of aerosol counts were significantly lower with the exhaust vents on with a statistically significant value. The utility of this observation, although of great value as a potential for reducing infectious aerosol particle

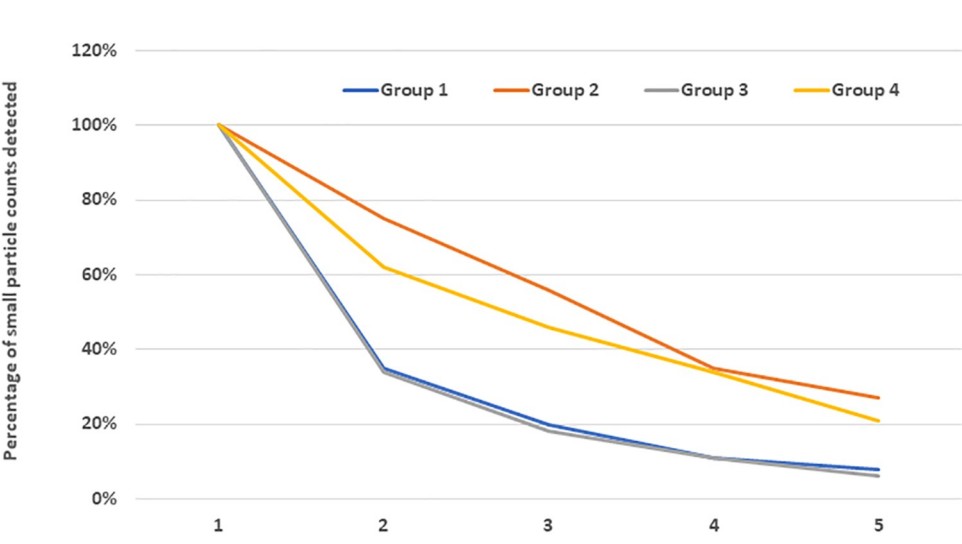

**Fig 6. Comparison of aerosol clearance between different groups from baseline to 120 mins.**

| Anova: single factor | | | | | | |
|---|---|---|---|---|---|---|
| Summary | | | | | | |
| Groups | Count | Sum | Average | Variance | | |
| Group 1 | 10 | 157804 | 15780.4 | 24378425 | | |
| Group 2 | 10 | 112254 | 11225.4 | 2069010 | | |
| Group 3 | 10 | 178953 | 17895.3 | 19553150 | | |
| Group 4 | 10 | 124950 | 12495 | 4437213 | | |
| | | | | | | |
| | | | | | | |
| Anova | | | | | | |
| Source of variation | SS | df | MS | F | p-value | F crit |
| Between Groups | 2.78E+08 | 3 | 92731142 | 7.354099 | 0.000575 | 2.866266 |
| Within Groups | 4.54E+08 | 36 | 12609450 | | | |
| Total | 7.32E+08 | 39 | | | | |

**Fig 7. One-way analysis of variance to determine statistical significance in the difference in means of air particles cleared in each group.**

number in a confined space in the home setting, may have its limitations because of variability in findings that depend on the specifications of kitchen vents with fans of different power in different homes. The effectiveness of the negative pressure generated in the space and subsequent air exchange that follows also depends on the size of the room, the distance of the seating area from the kitchen vent, the number of people in the room, and the location of the vent. The assessment of different types and models of kitchen vents in different homes and evaluation in closed or open floor plans was limited during the pandemic but can be undertaken in the future. Most kitchen vents come with specifications of CFM for the exhaust fan which can be used to calculate ACH for a given room. It is also possible that biological aerosols behave differently than artificially generated aerosols. Regardless, an absolute reduction in aerosol count could prove to be beneficial when ventilation with open windows or fans is limited by cold temperatures during winter when the transmission of viruses is higher. The transmission of bioaerosols is dependent on several factors and accurate replication could be a limitation of this home-based observational study as well and urgent studies are needed to further characterize transmission of infectious aerosols in this setting.

## Conclusions

This observational study suggests that ventilation in common rooms can lower the aerosol concentrations in the air and possibly lower the risk of contracting respiratory infections like COVID-19. Home exhaust systems like kitchen vents can be used beyond their original purpose of improving indoor air quality by clearing kitchen cooking fumes. Extended use during and after times when people gather together for meals in a kitchen or dining area could lead to enough air exchange to justify its use as a simple, yet effective, air exchange system in a home setting. Perhaps a multimodality approach to increasing ventilation and air exchange in homes

with more liberal and prolonged use of kitchen exhaust vents with or without open windows may be of value in reducing aerosol numbers in the home setting and mitigating the spread of respiratory infections. With global vaccinations lagging behind and new variants causing deadly surges, every modality that can reduce the transmission of the SARS-CoV-2 virus needs to be urgently implemented. Increasing public awareness that exhaust systems, that may already exist in homes, reduce aerosol counts effectively and liberal use around the time of indoor gatherings may reduce the transmission of the SARS-CoV-2 virus and other respiratory infections.

## Supporting information

**S1 Data.**
(DOCX)

## Acknowledgments

I would like to thank Leigh Thompson, Vonita Giddings, and Dr. David Headly for their support and help in proofreading my paper.

## Author Contributions

**Conceptualization:** Ashwin Johri.

**Data curation:** Ashwin Johri.

**Formal analysis:** Ashwin Johri.

**Investigation:** Ashwin Johri.

**Methodology:** Ashwin Johri.

**Validation:** Ashwin Johri.

**Writing – original draft:** Ashwin Johri.

**Writing – review & editing:** Ashwin Johri.

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
