## [Decision Letter · Decision Letter 0]

16 Aug 2021

PONE-D-21-23876

The Effect of Increasing Indoor Ventilation on Artificially Generated Aerosol Particle Counts

PLOS ONE

Dear Dr. Johri,

Thank you for submitting your manuscript to PLOS ONE. After careful consideration, we feel that it has merit but does not fully meet PLOS ONE’s publication criteria as it currently stands. Therefore, we invite you to submit a revised version of the manuscript that addresses the points raised during the review process. In particular, the reviewer requests that you update the references used in your manuscript.

We look forward to receiving your revised manuscript.

Kind regards,

David M. Ojcius

Academic Editor

PLOS ONE

Journal Requirements:

3. Please upload a copy of Figure 4, to which you refer in your text on page 5. If the figure is no longer to be included as part of the submission please remove all reference to it within the text.

4. We note that Figures 2 and 3 in your submission contain copyrighted images. All PLOS content is published under the Creative Commons Attribution License (CC BY 4.0), which means that the manuscript, images, and Supporting Information files will be freely available online, and any third party is permitted to access, download, copy, distribute, and use these materials in any way, even commercially, with proper attribution. For more information, see our copyright guidelines: http://journals.plos.org/plosone/s/licenses-and-copyright.

a. You may seek permission from the original copyright holder of Figures 2 and 3 to publish the content specifically under the CC BY 4.0 license. 

5. We note you have included a table to which you do not refer in the text of your manuscript. Please ensure that you refer to Tables 1 and 2 in your text; if accepted, production will need this reference to link the reader to the Table.

Reviewers' comments:

Reviewer's Responses to Questions

**Comments to the Author**

1. Is the manuscript technically sound, and do the data support the conclusions?

Reviewer #1: Yes

2. Has the statistical analysis been performed appropriately and rigorously? 

Reviewer #1: Yes

3. Have the authors made all data underlying the findings in their manuscript fully available?

Reviewer #1: Yes

4. Is the manuscript presented in an intelligible fashion and written in standard English?

Reviewer #1: Yes

5. Review Comments to the Author

Reviewer #1: 1) In the first sentences of the introduction, the authors should update the figures on covid-19 (number of infection and deaths in the world).

2) In introduction, I suggest to cite 3 studies concerning: 1) the high rate of viral spread within households and indoor place; 2) the mechanisms of airborne diffusion and implications for workers like healthcare professionals: Chirico F, Nucera G, Sacco A, Magnavita N. Proper respirators use is crucial for protecting both emergency first aid responder and casualty from COVID-19 and airborne-transmitted infections. Adv Respir Med. 2021;89(1):99-100. doi: 10.5603/ARM.a2021.0028. Chirico F, Sacco A, Bragazzi NL, Magnavita N. Can Air-Conditioning Systems Contribute to the Spread of SARS/MERS/COVID-19 infection? Insights from a Rapid Review of the Literature. Int J Environ Res Public Health. 2020, 17(17), 6052; https://doi.org/10.3390/ijerph17176052. Chirico F, Nucera G, Magnavita N. SARS-CoV-2 engines and containment strategy to tackle COVID-19 in Italy. Science. e-letter. 29 October 2020. Available from: https://science.sciencemag.org/content/370/6515/406/tab-e-letters.

3) The authors should separate results from discussion.

4) In discussion the authors should add something about practical implications of their experiment, study's limitations and a short conclusion.

5) more references are needed.

6. PLOS authors have the option to publish the peer review history of their article (what does this mean?). If published, this will include your full peer review and any attached files.

Reviewer #1: No

---

## [Author Response · Author response to Decision Letter 0]

26 Aug 2021

Dear Editor, 

Thank you for your response and for providing the opportunity to resubmit my manuscript. The comments that I have received have definitely strengthened my manuscript and I am grateful for this input. 

Please see below my point-by-point responses and related manuscript changes, and please let me know if these address your and the reviewers’ comments.

Journal Requirements:

1. Please ensure that your manuscript meets PLOS One’s requirements, including those for file naming

Response: The manuscript was modified as per the stated formatting guidelines

2. In your Data Availability statement, you have not specified where the minimal data set underlying the results described in your manuscript can be found

Response: A separate file with raw data set is uploaded under Supporting Information files

3. Please upload a copy of Figure 4, to which you refer in your text on page 5.

Response: The reference to Figure 4 has been removed

4. We note that Figures 2 and 3 in your submission contain copyrighted images.

Response: Figures 2 and 3 have been removed from the submission

5. We note that you have included a table to which you do not refer in the text of your manuscript. Please ensure that you refer to Tables 1 and 2 in your text; if accepted, production will need this reference to link the reader to the table

Response: Reference to tables 1 and 2 have been included in the text of the manuscript

Reviewer’s comments to the Author

Reviewer # 1

1. In the first sentences of the introduction, the authors should update the figures on covid 19 (number of infection and deaths in the world)

Response: The number of infections and deaths have been updated based on latest data Johns Hopkins COVID 19 coronavirus resource center from August 24, 2021

2. In the introduction, I suggest to cite 3 studies:

Response: The 3 recommended citations have been included as references 4, 11 and 20

3. The authors should separate results from discussion

Response: The results have been separated from the discussion

4. In the discussion the authors should add something about practical implications of their experiment, studies limitations and short conclusion

Response: The above mentioned segments have been separated and modified as per Reviewer 1’s suggestions. The practical implication of a simple method to improve indoor ventilation with kitchen exhaust vents has been included in the text. The limitations of the study which were included in the body of the text have been outlined with more clarity. A short conclusion has been added. 

5. More references are needed

Response: The following references have been added in appropriate places in the manuscript: 3, 4, 5, 11, 13, 17, 19, 20

Please also note that in order to reformat and make the recommended changes in the context of the paper, other unrecommended changes were necessary. All changes are shown in the marked copy of the manuscript. The digital files were processed through Pace and corrected version is uploaded.

I would like to thank you again for your time and consideration and I would be happy to respond to any other queries should they arise. 

Best regards,

Ashwin Johri

---

## [Editor Report · Decision Letter 1]

27 Sep 2021

The Effect of Increasing Indoor Ventilation on Artificially Generated Aerosol Particle Counts

PONE-D-21-23876R1

Dear Dr. Johri,

We’re pleased to inform you that your manuscript has been judged scientifically suitable for publication and will be formally accepted for publication once it meets all outstanding technical requirements.

Kind regards,

David M. Ojcius

Academic Editor

PLOS ONE
---

## [Editor Report · Acceptance letter]

30 Sep 2021

PONE-D-21-23876R1 

The effect of increasing indoor ventilation on artificially generated aerosol particle counts 

Dear Dr. Johri:

I'm pleased to inform you that your manuscript has been deemed suitable for publication in PLOS ONE. Congratulations! Your manuscript is now with our production department. 

Kind regards, 

on behalf of

Dr. David M. Ojcius 

Academic Editor

PLOS ONE